# Random walk informed heterogeneity detection reveals how the lymph node conduit network influences T cells collective exploration behavior

**Solène Song**[1]*, **Malek Senoussi**[1], **Paul Escande**[2], **Paul Villoutreix**[1¤]*

**1** Aix Marseille Univ, Université de Toulon, CNRS, LIS Turing Centre for Living Systems, Marseille, France,
**2** Aix Marseille Univ, CNRS, Centrale Marseille, I2M, Marseille, France

¤Current address: Aix Marseille Univ, INSERM, Marseille Medical Genetics, Marseille, France
* solene.song@univ-amu.fr (SS); paul.villoutreix@univ-amu.fr (PV)

## Abstract

Random walks on networks are widely used to model stochastic processes such as search strategies, transportation problems or disease propagation. A prominent example of such process is the dynamics of naive T cells within the lymph node while they are scanning for antigens. The observed T cells trajectories in small sub-volumes of the lymph node are well modeled as a random walk and they have been shown to follow the lymphatic conduit network as substrate for migration. One can then ask how does the connectivity patterns of the lymph node conduit network affect the T cells collective exploration behavior. In particular, does the network display properties that are uniform across the whole volume of the lymph node or can we distinguish some *heterogeneities*? We propose a workflow to accurately and efficiently define and compute these quantities on large networks, which enables us to characterize heterogeneities within a very large published dataset of Lymph Node Conduit Network. To establish the significance of our results, we compared the results obtained on the lymph node to null models of varying complexity. We identified significantly heterogeneous regions characterized as "remote regions" at the poles and next to the medulla, while a large portion of the network promotes uniform exploration by T cells.

## Author summary

Lymph nodes are organs in which actors of the immune system meet. In particular, the encounter between the naive T cells and their specific antigens occurs in lymph nodes. This event triggers the adaptive immune response. T cells movement has been shown to be well described as a random walk, at least when they are measured on small sub-volumes of the lymph node. In parallel, it was shown that T-cells migrate following the lymphatic conduit network that span the lymph nodes. In this study, we ask, how does the connectivity pattern of the conduit network on which T cells move influences their collective exploration behavior? Are there regions in the lymph node conduit network which have

**Data Availability Statement:** The code sufficient to reproduce our study can be found on the following github repository https://github.com/SoleneS/RW_

lymph_node and the additional data (approximated spectral decompositions of the transition matrices) https://zenodo.org/record/7681900#.Y 8UOS8w1hF.

**Funding:** SS, MS and PV were employees of Aix-Marseille University and funded by the "Investissements d'Avenir" French Government program managed by the French National Research Agency (ANR-16-CONV-0001) and from Excellence Initiative of Aix-Marseille University - A*MIDEX. PE was an employee of CNRS. The funders had no role in study design, data collection and analysis, decision to publish, or preparation of the manuscript.

**Competing interests:** The authors have declared that no competing interests exist.

distinct random walk related properties? The topological reconstruction of the lymph node conduit network was recently made available. The network is very large (about 200 000 nodes) and appears very regular, with most nodes being connected to three neighbours. We propose a workflow to detect heterogeneities in such large and quasi-regular networks, building on random walk on network tools, and the measure of two features which we interpret using a series of generated null models for comparison. We show that the lymph node conduit network displays remotely accessible regions at both poles and near medulla, with however most of the network promoting uniform exploration.

## Introduction

Random walks on networks are a widely used model to describe search strategies [1, 2], transportation problems [3], transmission in epidemiology [4, 5] or diffusion of information [6, 7]. In this model, random walkers hop from node to node while choosing randomly the edges on which to move. The structure of the underlying network, such as its degree distribution and connectivity pattern, will thus determine how the random walk evolves over time. Can the connectivity pattern favour the exploration of some nodes over others? We will test the hypothesis that the network is heterogeneous in the sense that regions can be distinguished using their connectivity properties, and that this heterogeneity has an influence on the collective exploration behavior on the network. We aim to define a measure of such heterogeneity that is applicable to large networks.

Indeed, the question of the influence of network connectivity on random walker behavior is raised in the case of biological transportation networks [8]. Among such networks, the lymph node conduit network (**LNCN**) offers a prominent example of a large network whose structure can affect its function [9]. The lymph nodes, among other functions, are hubs along the lymphatic system in which T cells encounter dendritic cells upon an infection. The dendritic cells bring the virus' antigen to the lymph node. Only a small subset of the naive T cells able to react to a given antigen (one out of 1 000 000). When these relevant naive T cells encounter the dendritic cells, they proliferate and the specific immune response starts. This crucial encounter arises after a search phase [10]: the dendritic cells stay still at an unknown location, and the naive T cells scan the lymph node for dendritic cells to test their specificity. The trajectories of T cells on sub-volumes of the lymph nodes have shown different types of random walk behaviours [10]: brownian [11], Lévy-type [12] or, more recently, correlated random walk [13]. These behaviours might arise as a combination of different mechanisms: internal cell reorganizations, chemical cues, or the fact that T cells follow a network, the **LNCN**, which is a network of pipes conveying lymph that span the lymph nodes, as support for their migration [14]. In this paper, we focus on this last aspect of the T cell behaviour, so that without additional hypothesis, T cells movement is modeled as a random walk on the **LNCN**. Considering the topology of the **LNCN**, we ask if the network connectivity influences the collective exploration behavior by T cells by exhibiting specific heterogeneities.

Previous studies have shown that the network formed by the cells that cover the conduits, the FRC (fibloblastic reticular cells), has a small-world structure and robustness [15]. This network is surprisingly different from the network formed by the conduits themselves (**LNCN**) which exhibits very different properties. These characteristics were measured on a slice of a mouse lymph node. Only recently, for the first time, the whole conduit network (**LNCN**) was imaged by Kelch et al. [9], providing a new and unique opportunity to map the connectivity properties across the whole T cell zone. The authors observed that there is a higher density of

nodes on the periphery on the network and lower density at the core. Focusing on the guiding function of the conduit network (**LNCN**), the authors performed an agent-based model to simulate T cells migration which concluded that immune cells would have similar behaviors in both regions. Similarly, and in the absence of an available complete T-cell zone reconstruction of the FRC network, we will consider the **LNCN** as the substrate of migration of the T cells, and not the FRC network, and thus base our analysis on the **LNCN** topological reconstruction from Kelch et al.

To answer the question of whether the network topology biases the search, the global mean first passage time (GMFPT) [16] might appear at first to be sufficient to detect which nodes are found by random walkers first. However this measure is local (node by node) and does not allow to delimit whole distinct regions in the network. Moreover, it is not computable for large networks since it requires the computation of all return probabilities for all nodes at all time steps. Here, instead, we propose to assess the level of heterogeneity of the network by comparing random walk related properties between regions, i.e. communities of nodes, of the network. If these features are similar in all regions, the network is homogeneous. On the contrary, if some regions have features differing considerably from the others, the network displays heterogeneity. To devise random walk interpretable communities, we compute clusters in the so-called diffusion space [17, 18]. These communities can be interpreted as groups of nodes which are highly connected by random walks paths, which means that there are many short paths connecting the nodes [19]. Additionally, random walkers departing from nodes of a same community have correlated probability of presence fields over time. These communities can be defined at different resolutions by tuning two parameters. First, one can vary the length of the random walk to be considered, i.e. the number of time steps after which the probabilities are calculated, and second, the number of communities which encodes the resolution at which we observe the network. Furthermore, the workflow is tractable for large networks.

The first question is: Do the diffusion communities form compact groups of neighbouring nodes or are they on the contrary scattered across the network? The latter case implies that in some cases there is more chance to reach a node at the other end of the network than a node which is only a few edges apart. We call this property *spatial coherence*, which describes how the diffusion accessibility is correlated with the shortest paths lengths. This question is motivated by a seemingly contradiction in the description of the lymphatic network in the lymph node. Indeed, on the one hand, the conduit network that was described qualitatively as a mesh, [9], a loosely defined concept that suggests high spatial coherence. On the other hand, the small-world property of the FRC network [20] suggests that there are shortcuts between otherwise distant nodes of the network [21]. To answer this question, we introduce the Cheeger mixing index as a measure of *spatial coherence*. We compare the average value of the Cheeger mixing index of the **LNCN** with a series of null models to estimate if this value is high or low. Using a Voronoi tesselation of similar size and mean degree as the **LNCN** as a null model for minimal Cheeger mixing index, we show that the Cheeger mixing index increases progressively as we rewire an increasing number of edges from the initial network. The Cheeger mixing value for the **LNCN** is very low, barely higher than the tesselation network showing that it displays high *spatial coherence*.

The second question is: is the network heterogeneous? This will be assessed by measuring the mean entry and exit probabilities of diffusion communities and assessing their variability. To answer this question, we measure the mean entry and exit probability at relaxation time, a reference time that can be found in all networks. We compare the levels of variation of the values between communities of the **LNCN** with various null models of the same size and mean degree and we conclude that the **LNCN** is significantly heterogeneous. We locate a few communities with lower mean entry and exit probabilities at the time scale of the exploration time

of the T cells. These communities are at the extremities of the longest axis and one near the medulla which is where the T cells exit the lymph node. On the contrary, the rest of the **LNCN**, appears homogeneous, promoting an uniform exploration by random walkers.

In summary, the value of this study is to build on the interpretability of the diffusion space coordinates [17–19, 22] and its associated approximation [22] to propose a workflow to detect heterogeneity in networks which are large, and which appear almost uniform in terms of degree distribution, such as the **LNCN**. The workflow succeeds at characterizing the network as *spatially coherent* and to locate specifically "remote" regions in the **LNCN**, by comparing the values of the indicators we propose between the **LNCN** and a series of null models.

## Materials and methods

In this section, we first describe our workflow, which consists in defining diffusion communities within the network, and to measure for each community (i) the Cheeger mixing index to characterize the *spatial coherence* of the network and (ii) the mean entry and exit probabilities to detect specific regions that are more accessible or remote. Then, we introduce the networks that are analyzed and compared using this workflow. The networks consist of the **LNCN** which is the biological network of interest, and 7 other generated networks of similar sizes and same mean degree as the **LNCN** as null models to help interpret the measured features.

### Workflow

We consider only the connectivity information of the network under consideration and ignore the spatial coordinates and edge lengths. Therefore, we address a discrete-time random walk on an unweighted, undirected network. The dynamics of a random walk on a network can be derived analytically. We use previously defined *diffusion coordinates*, based on the definition of a random walk [17, 19], to compute community detection. The workflow is illustrated on Fig 1.

**Random walk on a network.** Let $\mathcal{N} = (\mathcal{V}, \mathcal{E})$ be a spatial network where $\mathcal{V} \in \mathbb{R}^{3 \times N}$ is the set of N nodes and $\mathcal{E} = \{(i, j) \in [1, N]^2\}$ is the set of E edges. The connectivity information is encoded in the adjacency matrix $A$ which is defined as $A_{ij} = 1$ if $(i, j) \in \mathcal{E}$ and $A_{ij} = 0$ otherwise. We consider a random walker following a Markovian process on the network. At each time step, the walker chooses with equal probability to jump to one of its adjacent nodes. The transition matrix $T = D^{-1}A$ (where $D = diag(d_i)$ is the diagonal degree matrix with $d_i$ the degree of node $i$) encodes the probabilities of transition from one node to another in one time step $T_{ij} = p(j, t = 1|i)$.

**Diffusion coordinates.** The diffusion coordinates allow to embed each node of the network in an interpretable Euclidean space. These distances depend on the random walk on the network, thus the embedding is called the *diffusion space*. [22]. The diffusion coordinates are computed by using the spectral decomposition of the transition matrix T with left and right eigenvectors [19, 22, 23] $T = \Psi \Lambda \Phi^T$ (the derivation is in S1 Text), and for any time step $t = q$, $T^q = \Psi \Lambda^q \Phi^T$. The diffusion coordinates for a given time $t = q$ are given by the rows of the left part of the spectral decomposition: $X(q) = \Psi \Lambda^q$. The Euclidean distances in this embedding, also called diffusion distances are interpreted as a measure of the correlation of random walks departing from $i_0$ and $i_1$, with $(i_0, i_1) \in [1, N]^2$:

$$D_t^2(i_0, i_1) = \|p(j, t|i_0) - p(j, t|i_1)\|_w^2 \tag{1}$$

$$= \sum_j (p(j, t|i_0) - p(j, t|i_1))^2 w(j) \tag{2}$$

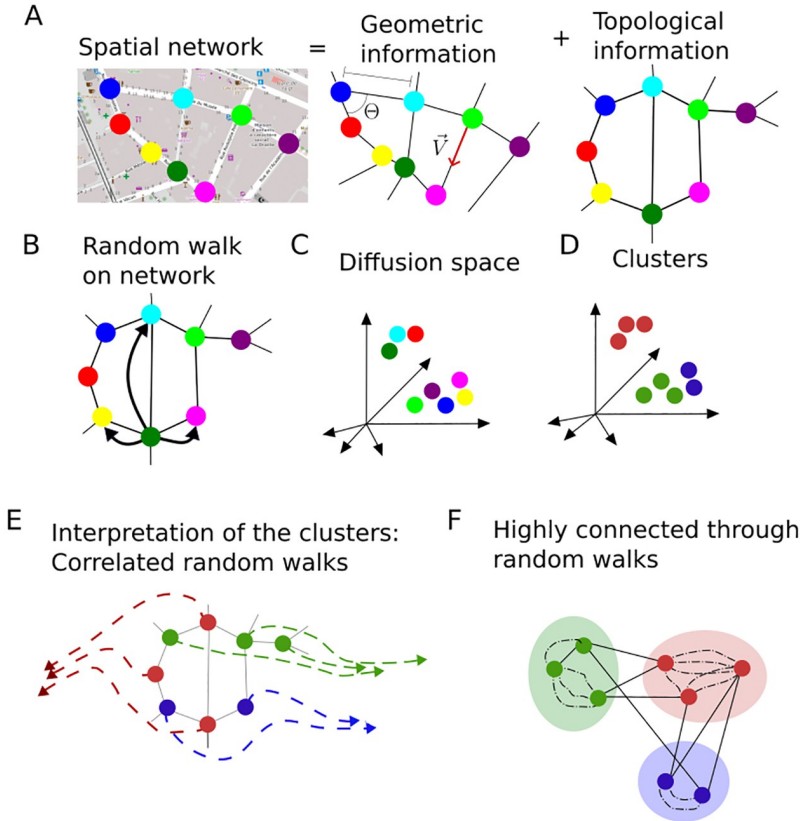

**Fig 1.** Description of the diffusion community detection workflow A: Decoupling of a spatial network into geometric and topological information. Credit for the base layer city map that illustrates the spatial network: Rue d'Aubagne, Marseille, France OpenStreetMap B: We consider a random walker on the topological network C: Nodes of the network are embedded into the diffusion space D: Diffusion communities are clusters made in the diffusion space E: These communities are interpreted as: random walkers departing from nodes of the same community follow correlated probability of presence fields F: Nodes are close in the diffusion space if they are highly connected through random walks. Dash lines represent random walks trajectories through nodes not represented on the sketch.

where $w(j) = 1/\phi_{j0}$, with $\phi_{j0} = \frac{d_j}{2E}$ [22]. Furthermore, nodes are close in the diffusion space if random walkers are likely to travel from one to the other through short paths. For large networks, the spectral decomposition of the transition matrix is computationally impossible. We therefore use an approximation.

**Approximation.**    For large number of nodes, such as for the **LNCN**, which contains approximately 200,000 nodes, the computation of $\Psi$ and $\Phi$ requires the diagonalization of the symmetrized transition matrix $T_s = D^{\frac{1}{2}}TD^{-\frac{1}{2}}$, which cannot be completed fully for large $N$. The matrix $T$ and the diffusion coordinates $X(q)$ can be respectively approximated by the truncated spectral decomposition $\hat{T}^t_K = \Psi_K \Lambda^t_K \Phi^T_K$ where $\Psi_K = (\psi_0, .., \psi_K)$, $\Phi_K = (\phi_0, .., \phi_K)$ and $\Lambda_K = \text{diag}(\lambda_0, ..., \lambda_K)$, and $X_k(q) = \Psi_k \Lambda^q_k$. The diffusion distance is then $D^2_t(i_0, i_1) = \sum_{k=1}^{K} \lambda_k^{2t} (\psi_k(i_0) - \psi_k(i_1))^2$. The numeric calculation and its algorithmic complexity are detailed in S2 Text. The accuracy of this approximation increases with increasing $t$ or when $K$ increases. In this paper, $K$ was restricted to $K = 2000$ for all networks to keep spectral decomposition computation time acceptable (11–13 hours). In the case of the **LNCN** the smallest $t$ we consider suffers from an error of less than 0.1%. The method for error calculation is detailed in S3 Text and the error for the **LNCN** with respect to $t$ is shown in S1 Fig.

**Community detection and relaxation time.** Using the diffusion coordinates, the nodes of the network can be seen as a point cloud. To identify subset of nodes having similar properties, we computed communities using the k-means algorithm [24] in the diffusion coordinate space. Since the diffusion coordinate are dependent of the time $t$, we chose to always use a reference time in the dynamics of the random walk, $\tau$ the relaxation time. The relaxation time is the time at which the difference between the probability field and the stationary field is reduced significantly by a constant factor. The relaxation time differs from one network to the other, however providing a comparable time scale. This time scale is defined by the ratio between the magnitude of the second largest eigenvalue of the transition matrix $|\lambda_1|$ and $\lambda_0 = 1$, $\tau = \frac{1}{1-\lambda_1}$. The derivation is in S4 Text and the values for all the considered networks are summarized in Table 1. As for $k$, the number of clusters, we chose to use $k = 100$ for all networks. We show how this k-dependant community detection differs from state-of-the-art random walk based community detection method Infomap [7] in S2 Fig.

To analyze and understand the properties of the communities, we compute two features, the Cheeger mixing index, and the mean entry and exit probabilities. The Cheeger mixing index measures to what extent the nodes which belong to the same diffusion community, thus which are close in the diffusion space, form compact neighbourhoods in the network. The mean entry and exit probabilities are features that can be computed over time, and can be compared between communities to distinguish outlying communities with especially low or high values.

**Cheeger mixing index** The Cheeger mixing index, for the community $C$, is defined as

$h(C) = \frac{\sum_{i \in C} \sum_{j \in \bar{C}} A_{ij}}{\min(\sum_{i \in C} d_i, \sum_{i \in \bar{C}} d_i)}$ and measures the relative number of edges connecting a node of

$C$ and a node that does not belong to $C$. If it is large it means that the nodes that belong to the same diffusion community do not form contiguous groups. We call this value Cheeger mixing index because the minimal value over all possible sets of nodes instead of community C is known as the Cheeger constant [25]. This measure is similar to the participation coefficient defined in [26]. The Cheeger mixing index, measured on diffusion communities, can be interpreted as follows. The diffusion distance $D_t^2(i_0, i_1)$ between two nodes $i_0$ and $i_1$ expresses the distance between the two posterior distributions $p(j, t|i_0)$ and $p(j, t|i_1)$. If $i_0$ and $i_1$ belong to the same community (i.e. are close in terms of diffusion distance), random walkers starting from $i_0$ and $i_1$ have correlated walks. Additionally, $D_t^2(i_0, i_1)$ is small if there is a large number of short paths connecting $i_0$ and $i_1$ [19]. Thus, a community defines a set of nodes that are highly connected and from which random walks are correlated. The Cheeger mixing index of one community $C$ measures the proportion of edges between nodes of $C$ and nodes that do not belong to $C$ over the number of edges inside $C$. The mean Cheeger

**Table 1. Summary of the characteristics the considered networks.**

| Network | Nb of nodes | Nb of edges | Modularity | $\tau$ |
|---|---|---|---|---|
| LNCN | 192,386 | 274,906 | 0.95 (n = 75) | 17,797 |
| Random | 192,306 | 269,308 | 0.72 (n = 225) | 1 |
| PVor | 193,429 | 270,786 | 0.94 (n = 78) | 10,924 |
| HVor | 204,653 | 286,509 | 0.94 (n = 79) | 5,339 |
| HVor rwd. 2% | 204,653 | 286,509 | 0.90 (n = 108) | 152 |
| HVor rwd. 5% | 204,653 | 286,509 | 0.85 (n = 194) | 104 |
| HVor rwd 10% | 204,653 | 286,509 | 0.80 (n = 237) | 100 |
| HVor rwd 20% | 204,653 | 286,509 | 0.85 (n = 230) | 82 |

mixing index is the average over all communities of the Cheeger mixing index $\bar{h} = <h(C)>_C$. For a given network, if this number is high we expect the nodes belonging to a same community to be scattered across the networks: close neighbours (in terms of shortest path length) are not necessarily the most well connected at this time $t$ integrating on all possible paths. On the contrary if this number is low, the nodes belonging to the same community form a compact neighbourhood within the network, which means high spatial coherence.

**Mean entry and exit probabilities** For each community, the mean entry probability was computed as the sum of probabilities to arrive at any node that belong to the community departing from all the nodes that do not belong to the community, averaged over all the nodes that belong to the community, then over all the nodes that do not belong to the cluster. $< p_{in} >_C (t) = \frac{1}{n_C n_{\bar{C}}} \sum_{l \in C} \sum_{m \in \bar{C}} \sum_{k=1}^{K} \psi_k(m) \lambda_k^t \phi_k(l)$. Similarly, the mean exit probability was computed as the sum of probabilities to arrive at any node that does not belong the community departing from any node that belongs to the communities, averaged over all the nodes that belong to the community, then over all the nodes that don't belong to the cluster. $< p_{out} >_C (t) = \frac{1}{n_C n_{\bar{C}}} \sum_{l \in \bar{C}} \sum_{m \in C} \sum_{k=1}^{K} \psi_k(m) \lambda_k^t \phi_k(l)$. Note that (i) if a network is fully regular, $< p_{in} >_C (t) = < p_{out} >_C (t)$ for all $C$ and (ii) at infinite time $t$, $< p_{in} >_C (t)$ is proportional to the mean degree of the community, and $< p_{in} >_C (t)$ is proportional to the mean degree of the nodes that do not belong to the communities (derivation in S5 Text). At long time scale, these measures depend only on the degree distribution among the communities but their computation at shorter time scales prove useful to distinguish communities, especially when the networks are quasi-regular as shown in Results section.

## Datasets

**The lymph node conduit network (LNCN).**   The network's connectivity, published in [9], was extracted from a segmented 3D microscopy acquisition of a whole mouse popliteal lymph node conduit network (850 x 750 x 900 $\mu$m) obtained from microscopy data [9] in which the conduits are made fluorescent by injection of labelled molecular tracer into the lymphatic vessels. The conduit network is restricted to the T cell zone, after the medulla, subcapsular sinus and B cell follicles were removed by the authors Kelch et al. This is a 3D network made of 192,386 nodes and 274,906 edges. Most of its nodes are degree 3 (72%), and 99% of nodes having degree between 1 and 4 (see degree distribution in Fig 2A).

**Homogeneous Voronoi (HVor).**   The homogeneous Voronoi 3D was generated using the network generator from Python library Scipy [27] spatial.Voronoi. The input points coordinates were defined such as: (i) create a rectangular 3D grid of 31x31x31 nodes (ii) keep only nodes inside a sphere resulting in 5185 nodes (iii) introduce noise in the coordinates so that the grid structure is not perfect (if the node were perfectly aligned on a 3D grid, this would correspond to a limit case where Voronoi vertices are degree 6 instead of 4). Then, the edges were randomly removed until the mean degree is 2.8 like the **LNCN**. The resulting network contains 204,653 nodes and 286,509 edges.

**Polar Voronoi (PVor).**   The polar Voronoi 3D was generated using the network generator from Python library Scipy [27] spatial.Voronoi. The input points coordinates were defined such as: the union of two Gaussian distributions in 3D, one centered in (0,0,0) with standard deviation 5, with $N_1 = 10,000$ nodes, and the other one centered in (5,5,5) with standard deviation 1 with $N_2 = 18,752$. Then, edges were randomly removed until the mean degree is 2.8 like the **LNCN**. The resulting network contains 193,429 nodes and 270,786 edges.

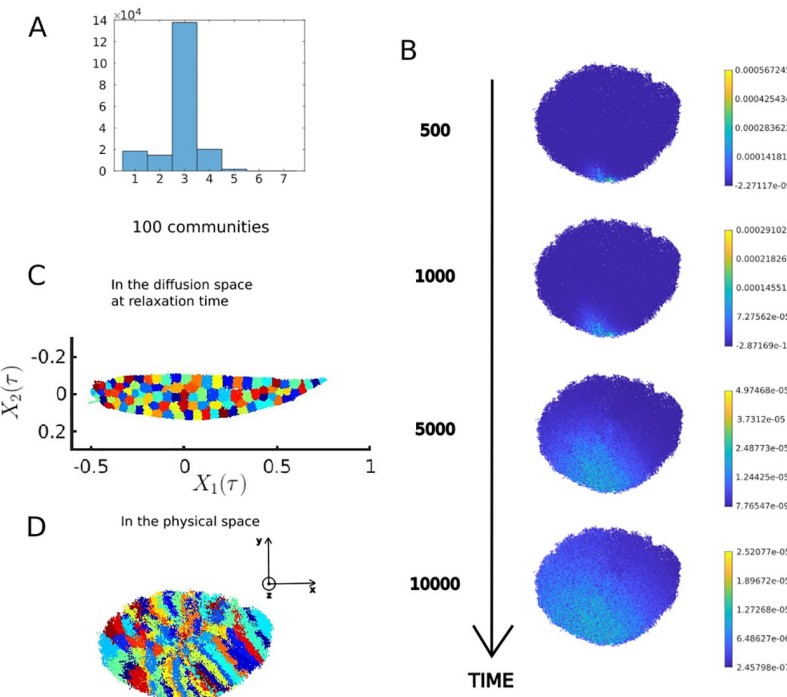

**Fig 2. LNCN features.** A: Degree distribution B: Probability of presence departing from one chosen node in time C: 100 diffusion communities at relaxation time $\tau$ shown in the diffusion space in the dimensions 1 and 2 (respectively 2nd and 3rd columns of $X(\tau) = \Psi\Lambda^{\tau}$, the first column for dimension 0 being filled with only value 1) D: The 100 diffusion communities shown in the physical space (right panel).

**Random network.** The random network was generated using configuration model [28, 29] from Python NetworkX library [30], using a degree sequence of a regular graph of degree 3 of same size as **LNCN**. Then, edges were randomly removed until the mean degree is 2.8 like the **LNCN**.

**Rewired Homogeneous Voronoi.** Departing from the **HVor**, we obtain the **HVor rewired 2%**, **HVor rewired 5%**, **HVor rewired 10%** and **HVor rewired 20%** respectively by double edges swapping [30] of 2%, 5%, 10% and 20% of edges.

## Results

### The LNCN despite quasi-regularity, shows anisotropy and heterogeneities

The **LNCN**, for which the topology was reconstructed by Kelch et al. [9], is a very large network. Overall the network is quite regular (more that 72% of nodes of degree 3) although there is a small amount of nodes of higher degree. Its size restricts the measures that can be made on the network because of the computational challenges, and its quasi-regularity gives a first impression of uniformity.

**Tracking random walk in the LNCN highlights the spatial organization of the network.** For any time $t$, the probability of presence of the random walker at time $t$ departing from a given node can be read as a term of the transition matrix elevated at the power $t$. However, the elevation at power $t$ of the transition matrix becomes intractable when $t$ grows higher and the matrix loses its sparsity. Thus, we apply the approximation described in the Methods section to the transition matrix. One can visualize and follow over time at low

computational cost the field of probability of presence of a random walker over time departing from a given node, thanks to the approximation of the transition matrix detailed in S2 Text. Fig 2B shows these fields departing from a given node from time 500 to 10,000 (for comparison, the relaxation time is reached at $\tau = 17,797$). The chosen node is the one with smallest y value. It is located at the center of the x-axis (longest axis). We see that the probability field in time is skewed towards the left part on the x-axis. The anisotropy of these fields hints towards a non uniform spatial organization of the network. Furthermore, the probability fields provide a way to explore the trajectories of random walkers departing from specific places chosen according to biological hypotheses without computing a Monte Carlo simulation.

**Description of the diffusion communities in the LNCN.** 100 diffusion communities were defined on the **LNCN** as described in the Methods section by using approximated diffusion coordinates at $t = \tau = 17797$. These diffusion communities are shown on Fig 2C. The cluster size varies from 272 to 4905 nodes and has an average value of 1924 nodes. The diffusion communities form elongated bands in the y-axis as shown in Fig 2D. Overall, the communities seem to form slices that are stacked along the x-axis. This suggests easier diffusion in the y and z-axes than in the x-axis.

The following sections will measure features on the communities and compare them to null networks in order to check the spatial coherence of these communities and the level of heterogeneity. The features colormaps of $< p_{in} >_C (\tau)$ and $< p_{out} >_C (\tau)$ as well as the Cheeger mixing values and mean degree for each community of the **LNCN** can be found in S3 Fig.

## The comparison of the Cheeger mixing index between LNCN and the null models shows that the LNCN is spatially coherent

The Cheeger mixing index averaged over all the diffusion communities, $\bar{h}$, like introduced in the Methods section quantifies to what extent the nodes which are close in the diffusion space are also close neighbours in terms of shortest paths in the network. We call this feature spatial coherence. This measure aims to clarify the structure of the **LNCN**. Indeed, as described in the introduction, the fact that the **LNCN** is called a mesh [9] suggests that we should find high spatial coherence. However, the FRC network that ensheath the conduits where shown to display small-world property [20]. The small-world property implies that the network minimizes the mean shortest path length between two nodes and maximizes the *clustering coefficient*, which refers to the propensity to form triangles, so that nodes that are connected to the same node tend to be also connected between them. As a consequence, there should be shortcut edges connecting distant nodes, thus low spatial coherence.

To illustrate how the Cheeger mixing index captures the spatial coherence of networks, we compute it on a series of networks made from increasing rewiring of the **HVor**. The **HVor** being a tesselation, it displays the lowest value of Cheeger mixing index. The more rewired edges there are, the more there are chances of shortcuts, and thus the more the Cheeger mixing value is high as shown in Fig 3. This seems to be a consistent definition for quantifying the spatial coherence of a network. The **LNCN** being between the **HVor** and the **HVor rewired 2%** shows that the **LNCN** is spatially coherent.

We have shown that the **LNCN** is spatially coherent, and we measured the heterogeneity in the **LNCN** across time, showing that its level of heterogeneity is significant compared to null models. In the following section we will locate spatially this heterogeneity, specifically at the time scale that is relevant for T cells scanning collective behavior in order to draw biological conclusions.

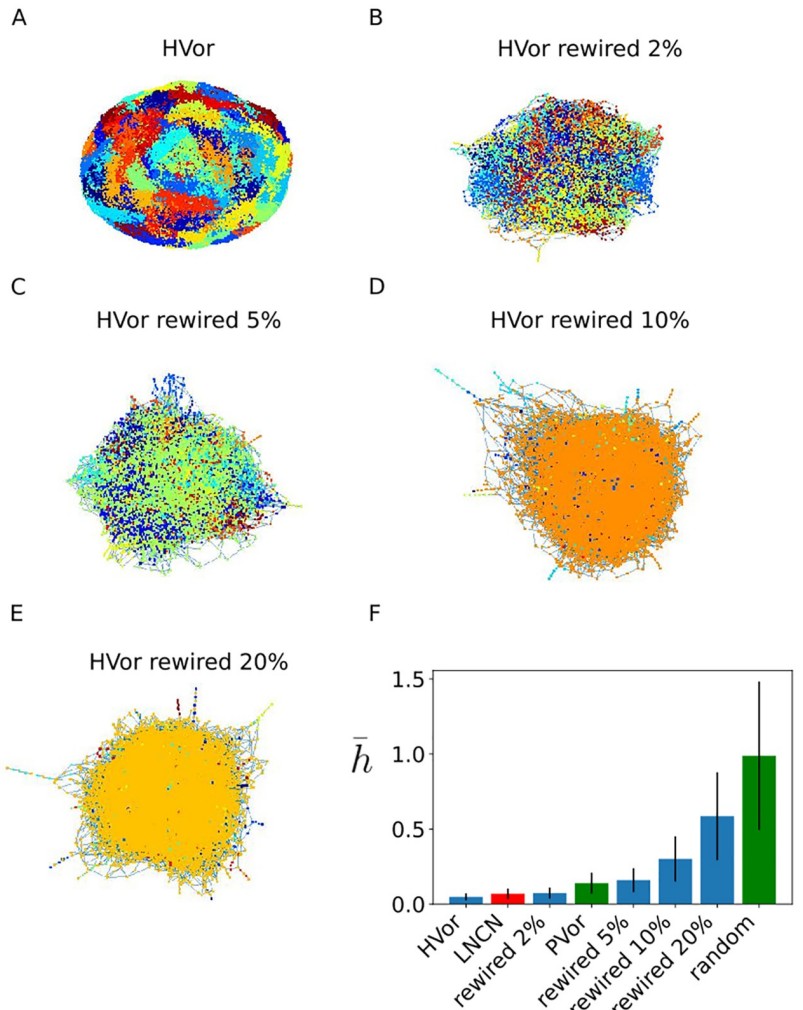

**Fig 3. The LNCN is spatially coherent.** A-E 100 diffusion communities for the **HVor**, **HVor rewired 2%**, **HVor rewired 5%**, **HVor rewired 10%**, **HVor rewired 20%**. Each color represent a community. In the **HVor** and **PVor**, there is respectively 95% and 99% of nodes that belong to a unique community among the 100 communities imposed by k-means algorithm F: The mean Cheeger index across all communities $\bar{h}$ for each null model. $\bar{h}$ increases as the rewiring percentage increases departing from the **HVor**. Compared with this series of null models, **LNCN** mean Cheeger mixing value is higher than the **HVor** and lower than the **HVor rewired 2%**.

## $< p_{in} >_C$ and $< p_{out} >_C$ comparison between the LNCN and null models shows that the LNCN is significantly heterogenous

We analyzed 3 different null networks of comparable sizes and mean degree in order to evaluate the level of heterogeneity of the **LNCN** by comparing the $< p_{in} >_C (t)$ and $< p_{out} >_C (t)$ variability to those of the null networks. The null networks were analyzed following the same workflow as the **LNCN**: we defined 100 communities by k-means clustering in the diffusion space. The time $t$ chosen to compute the diffusion coordinates is the relaxation time $\tau$. For each network, $< p_{in} >_C (t)$ and $< p_{out} >_C (t)$ are computed for each community at various times. For each time, the distribution of $< p_{in} >_C (t)$ and $< p_{out} >_C (t)$ for all 100 communities is displayed as boxplot in Fig 4A, 4B, 4C and 4D.

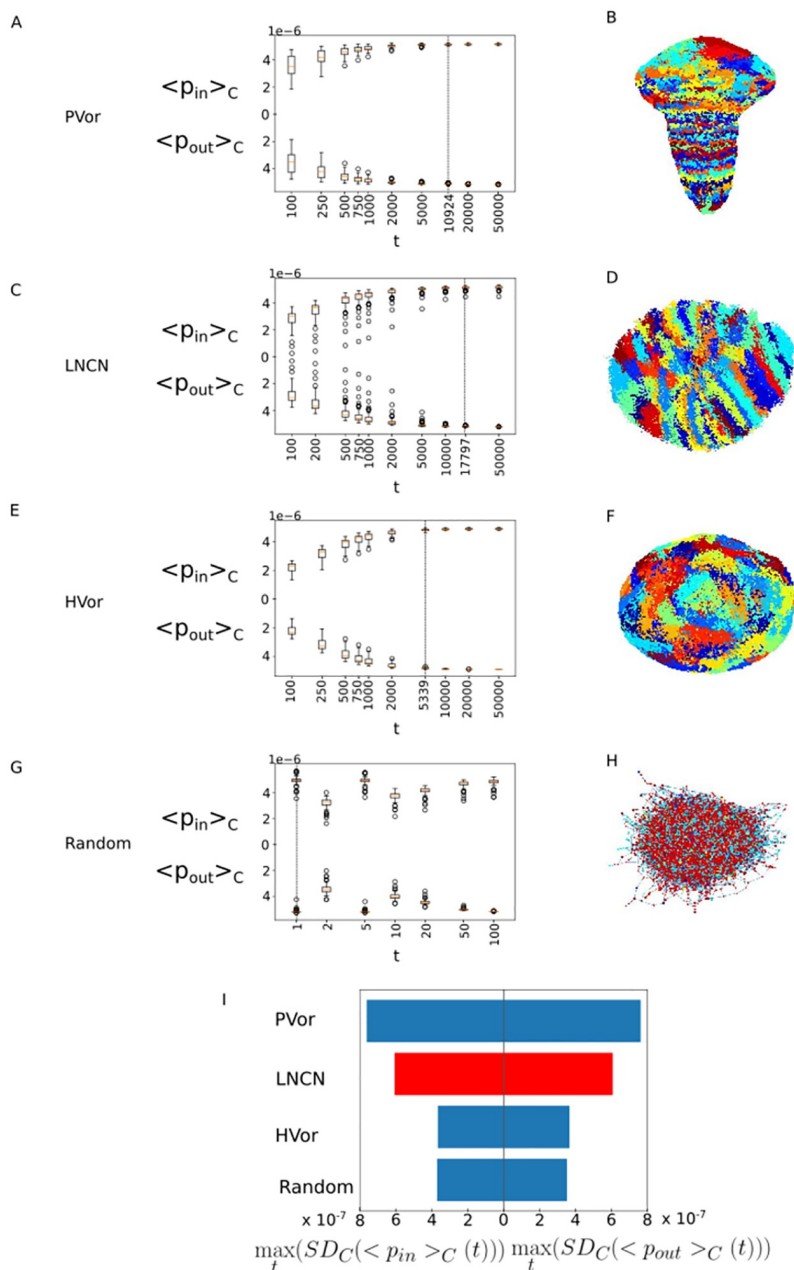

**Fig 4. The LNCN network displays a level of heterogeneity higher than the HCN and the Random network and lower than the PCN.** A,B,C,D: $< p_{in} >_C (t)$ and $< p_{out} >_C (t)$ box plots, respectively for the **LNCN,HVor**, **PVor** and **Random network**. The dash lines mark the relaxation times. E: The maximal standard deviation over all times of $< p_{in} >_C (t)$ and $< p_{out} >_C (t)$.

The diversity of values of $< p_{in} >_C (t)$ and $< p_{out} >_C (t)$ across the 100 communities is used as a proxy for heterogeneity across the communities. This diversity is computed as the standard deviation of $< p_{in} >_C (t)$ and $< p_{out} >_C (t)$ across communities. Since this depends on the random walk time considered, we take the maximal value over time. These maximal value of standard deviation of $< p_{in} >_C (t)$ and $< p_{out} >_C (t)$ across communities $\max_t(SD_C(< p_{in} >_C$

$(t)))$ and $\max_t(SD_C(<p_{out}>_C(t)))$ are recapitulated in Fig 4E. The higher these values are, the more heterogeneous the networks are.

The null networks are the **Random network**, the **HVor** and the **PVor**. The **Random network** has no structure so it is a first null model with minimal heterogeneity. The **LNCN** is more heterogeneous than the **Random network**. However, we know, from the previous section that the **LNCN** is very different from the **Random network** in terms of *spatial coherence*. So we also compare it with the **HVor** and **PVor**. The **LNCN** has higher heterogeneity than **HVor** but lower than **PVor**. We suppose that this is due to the fact that **HVor** was generated with spatially homogeneously distributed seeds whereas **PVor** which was generated with seeds with non-uniform spatial density, and this reflects into the network heterogeneity. In conclusion, the **LNCN** displays a level of heterogeneity which is significant, since it is higher than the **Random network** and **HVor**, and interestingly it is lower than the **PVor**, which indicates a polarity of the network. The diffusion communities and features colormaps ($<p_{in}>_C(\tau)$ and $<p_{out}>_C(\tau)$ as well as the Cheeger mixing values and mean degree) of the null models used in this section are in S3 Fig.

### The LNCN displays remote regions at its poles and next to the medulla, while being homogeneous on most of its total volume, promoting overall uniform collective exploration behavior

The biological scanning time corresponds to $t = 935$ steps. Indeed, the CD4+ T cells stay about 12h in the lymph node. With an estimation that the naive T-cells move with an average speed of $13\mu m.min^{-1}$ [31] and that the average length of an edge is $10\mu m$ [20], we can make the rough estimation that one time step represents 0.77 min. Thus, a 12h exploration time corresponds to $t = 935$ step time. We show in Fig 5A the values of $<p_{in}>_C$ and the $<p_{out}>_C$ for this time step. For this time step, the $<p_{in}>_C$ and the $<p_{out}>_C$ variation across communities is larger than at relaxation as shown in Fig 4A.

Heterogeneities are localized in restricted regions on the long axis extremities, the poles, and near the medulla. Fig 5B shows the values of $<p_{in}>_C(t = 935)$ and $<p_{out}>_C(t = 935)$ ranked from smallest to largest across all 100 communities. The heterogeneities that we describe are the 5 communities with smallest $<p_{in}>_C(t = 935)$ and $<p_{out}>_C(t = 935)$. We characterize these regions as *remote*, because a low value of $<p_{in}>$ indicates that it is more difficult to enter, and a low value of $<p_{out}>$ indicates that it is more difficult to get out of these regions.

We also computed the 5 communities with respectively the smallest and largest Cheeger mixing (Fig 5C and 5D). Among the 5 communities with smallest Cheeger mixing index, 4 of them are also part of the 5 communities with smallest $<p_{in}>_C(t = 935)$ and $<p_{out}>_C(t = 935)$. The 5 communities with largest Cheeger mixing index are communities which shapes are slices cutting the longest axis at its center.

Beside these regions that we identify with the extremal values of $<p_{in}>$, $<p_{out}>$ and Cheeger mixing index, we find that most of the network is homogeneous with similar values of these descriptors. Therefore there is no privileged regions for which random walkers would be attracted to or repelled from. This indicates that, despite variations in the connectivity patterns, the overall network structure promotes an exhaustive exploration by a collection of random walkers that maximizes the volume of exploration.

## Discussion

The question that we addressed is to assess whether the network connectivity influences the collective exploration behavior of the T cells.

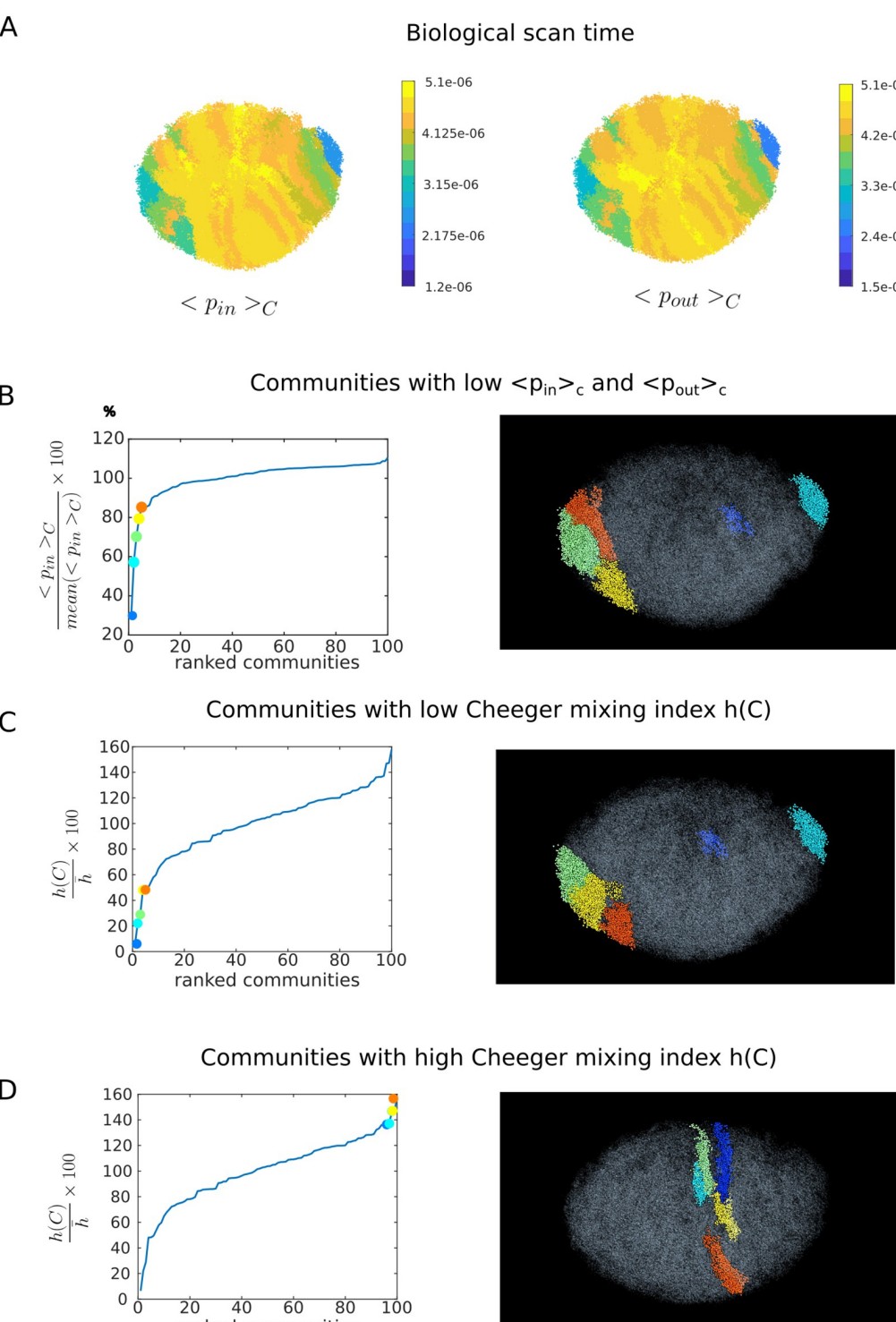

**Fig 5.** A: $< p_{in} >_C$ ($t = 935$) and $< p_{out} >_C$ ($t = 935$) for each community at biological scan time B: On the left panel, $< p_{in} >_C$ ($t = 935$) at biological scan time showed as percentage of the mean value across the 100 communities, shown for each of the 100 communities, ranked from smallest to largest value. The five communities with smallest values of $< p_{in} >_C$ are shown as colored dots. The same plot for $< p_{out} >_C$ is visually identical so we show only $< p_{in} >_C$. The same 5 communities have both lowest $< p_{in} >_C$ and $< p_{out} >_C$. On the right panel these five communities locations are shown with the same colors. C,D: On the left panels, Cheeger mixing value of each community represented as percentage of the mean value across the 100 communities, shown for the 100 communities, ranked from smallest to largest value. The five communities with smallest (C), and highest highest (D) values are shown as colored dots. On the right panels, these communities locations are shown with the same colors.

In summary, the **LNCN** promotes spatially continuous and overall uniform exploration by random walk. The low Cheeger mixing shows that similarly to a tesselation, in the **LNCN** the regions that are highly connected by short random walk paths are also close in the network in the sense of topological distance. Compared to the null models that we explored, the level of heterogeneity is significant. We distinguish a small set of communities which have lower $< p_{in} >_C$ and $< p_{out} >_C$ as well as small Cheeger mixing values. They are restricted to extremities along the long axis and close to the medulla. Since the entry and exit probabilities are low, they are not drawing T cells faster than other regions and thus are not particularly privileged location for the dendritic cells to sit to be found by the T cells. The rest of the network (most of the volume) is homogeneous, thus overall the network promotes uniform exploration by a large population of random walkers. In this respect, our finding that the **LNCN** promotes overall uniform exploration is compatible with an optimal search strategy that would aim at maximizing the volume crossed by random walker.

We have restricted the analysis to the topology of the network, and we have not included the geometric information: edges lengths and angles. For a more complete view of the network, including some assumptions on the preferential angles can be described by a weighted network, so that edges which are for instance most aligned with the one the walker comes from have a higher probability to be chosen. Taking into account the edges lengths, and subsequent traveling times, requires to address continuous time random walk to account for the fact that some edges take more time than others to be crossed, which can also be formulated as: the waiting time between two jumps will be long if the second jump involves crossing a long edge.

Our conclusions about the exploration behavior of the T cells also suffers several limitations. The measures were done on one sample of lymph node. Including more samples would increase the robustness of the conclusions. Ideally, staining of new lymphatic conduit networks would be done simultaneously with the staining of the entry and exit locations of the lymph nodes, respectively the HEVs (high endothelial venules) which are part of the blood vascular network, and the efferent lymphatic vessels at the medulla. One would then be able to include inlet and outlet in the random walk in the network analysis. Another point that would need clarification is the radical mismatch between the FRC network and the conduit network characteristics. The FRC network, in which nodes and edges are respectively the nuclei and cellular protrusions of the cells that ensheath the conduits, based on a slice of lymph node, shows small world properties and a degree distribution far less regular than the conduit network. T-cells can be guided by FRC as they are producing IL7 survival factors [32] and adhesion molecules [33, 34], however, no complete reconstruction of the FRC network is available to date. Moreover, the FRC use the conduit network as a support, therefore the constraints induced by the topology of the conduit network would still apply when considering the T-cells migration on a network of FRC. Our workflow can easily be applied on the FRC network when the topological data of whole lymph node FRC network becomes available.

Finally, the conduit network has other functions than guiding the migration of the T cells. First, its structure is related to mechanical robustness of the lymph node [35]. Then, it is also a piping system in which lymph flows, conveying crucial immune system molecules such as antigens, inflammatory soluble mediators and cytokines across the lymph node. Thus its structure could also be analyzed from a hydrodynamics point of view, such as modeled in the cortex capillary network for example [36, 37]. We anticipate that our finding would be consistent with a problem of optimal transport within the conduit network. Both flowing inside the network and a random walk using the network as support should share similar characteristics as continuous and discrete flow models.

## Supporting information

**S1 Text. Detailed spectral decomposition of the transition matrix.**
(PDF)

**S2 Text. Numerical approximation of the spectral decomposition for the large networks.**
(PDF)

**S3 Text. Estimation of the error of the approximation of the spectral decomposition of the transition matrix.**
(PDF)

**S4 Text. Relaxation time calculation.**
(PDF)

**S5 Text. Features of $< p_{in} >_C$ and $< p_{out} >_C$.**
(PDF)

**S1 Fig. Relative error estimation of the approximated transition matrix for the LNCN.** Relative spectral norm between the exact transition matrix and the approximated one with truncation at k = 2000 first eigenvalues, decaying with time steps. Insert: log scale representation.
(PDF)

**S2 Fig. This figure shows how diffusion communities differ from Infomap [7], a state-of-the-art community detection method based on random walk.** A-D: small toy network that exemplifies the difference between diffusion communities and Infomap. In this network, Infomap (D) detects only one community, that encompasses all the nodes. In the workflow we propose, the diffusion communities are made after choosing k, the number of communities when applying the k-means algorithm on the diffusion coordinates, for $t = \tau = 1$. k is a parameter one needs to adjust, which allows to tune the resolution. From $k = 2$ to $k = 4$ (A-C) the diffusion communities are more and more precise. For $k = 4$, the communities illustrate well the interpretation of diffusion communities as groups of nodes from which random walkers follow close trajectories. E: Infomap algorithm applied on the **LNCN** yields 2 clusters, with code-length 5.75. Diffusion communities, as presented in this study allows to detect higher resolution communities by chosing $k = 100$.
(PDF)

**S3 Fig. Features colormaps for the diffusion communities in the null networks Random network, HVor, PVor, and for the LNCN.** For the null networks, the networks are represented as graphs, the embedding being the default layout of Matlab. On the contrary, the **LNCN** is represented with the 3D coordinates of its nodes. For each network, for each community we show the values as colormap of the mean degree $< d >_C$, the mean entry and exit probabilities $< p_{in} >_C (\tau)$ and $< p_{out} >_C (\tau)$ at relaxation time, as well as the Cheeger mixing value $< h >_C$.
(PDF)

## Acknowledgments

We thank Marc Bajénoff for initial discussions on the biological aspects of T cells migration in the lymph node. We thank Inken Kelch, Gib Bogle and Rod Dunbar for providing the lymph node conduits data, and fruitful discussions. We thank Romain Pousse and Stéphane Douady for fruitful discussions. We thank Anaïs Baudot, Anthony Baptista and Alain Barrat for their critical reading of the manuscript and helpful discussions. We thank Nicolas Levernier, Jean-

François Rupprecht and Tanguy Fardet for helpful discussions. We thank Guillaume Gay from the Multi-Engineering Platform of the Turing Center for Living Systems, Marseille, France, for his valuable technical support.

## Author Contributions

**Conceptualization:** Solène Song, Paul Villoutreix.

**Data curation:** Solène Song.

**Formal analysis:** Solène Song, Malek Senoussi, Paul Escande, Paul Villoutreix.

**Funding acquisition:** Paul Villoutreix.

**Investigation:** Solène Song, Malek Senoussi, Paul Escande.

**Project administration:** Paul Villoutreix.

**Software:** Solène Song, Malek Senoussi.

**Supervision:** Paul Villoutreix.

**Validation:** Solène Song.

**Visualization:** Solène Song.

**Writing – original draft:** Solène Song, Paul Villoutreix.

**Writing – review & editing:** Solène Song, Malek Senoussi, Paul Escande, Paul Villoutreix.

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
