## [Decision Letter · Decision Letter 0]

31 Dec 2022

Dear PhD Song,

Thank you very much for submitting your manuscript "Random walk informed community detection reveals heterogeneities in the lymph node conduits network" for consideration at PLOS Computational Biology.

As with all papers reviewed by the journal, your manuscript was reviewed by members of the editorial board and by several independent reviewers. In light of the reviews (below this email), we would like to invite the resubmission of a significantly-revised version that takes into account the reviewers' comments.

We cannot make any decision about publication until we have seen the revised manuscript and your response to the reviewers' comments. Your revised manuscript is also likely to be sent to reviewers for further evaluation.

Sincerely,

James R. Faeder

Academic Editor

PLOS Computational Biology

Mark Alber

Section Editor

PLOS Computational Biology

Reviewer's Responses to Questions

**Comments to the Authors:**

Reviewer #1: Song et al. develops a method to assess heterogeneous networks using the Cheeger mixing index. The authors then develop both 2D and 3D toy networks and use the Cheeger mixing index to compare features of network connectedness to data from staining of lymph nodes obtained from Kelch et al 2019.

The use of new quantitative measures to determine biological features of networks such as LNs is novel and interesting. However, there are several issues with the application of the Cheeger value both to model networks and biological networks. For example, while Fig 4A shows interesting differences between the HCN and PCN, the Cheeger mixing index (Fig 4F) shows similar values for HCN and PCN. Although the Cheeger value can distinguish between random networks and HCN and PCN, it’s not clear how useful this measure is likely to be to actually distinguish complex networks such as biological networks of LNs. Either larger planar networks should be tested to show that Cheeger mixing value can be useful to distinguish or discussion of how Cheeger is meaningful needs to be better explained. Otherwise, it’s not clear how Cheeger value is useful.

An additional problem with whether Cheeger value can distinguish between biologically distinct networks also came with the 3D toy networks. In Fig 5J, the Cheeger value looks similar between LNs, PVor, and HVor. Thus, it is difficult to assess whether Cheeger can shed any new light on network properties, either toy networks or biological networks. As the manuscript stands, it is not clear Cheeger value will give meaningful interpretation to 3D networks.

Of even more significant concern is the applicability of the Cheeger mixing value to biological data such as lymph nodes. To utilize biological data, reproducibility must be considered. The authors already point out a significant and problematic limitation of the use of Kelch et al. data in that only one lymph node is stained. To base quantitative measurements on a single data point is not acceptable as reliable biological data.

Furthermore, there are multiple types of stainings utilized in Kelch et al, including WGA staining which stains the full lymph node, including B cell follicle, germinal centers, T cell zone, and medullary regions. These areas are distinguished with antibody staining but it is not clear from Song et al. whether this manuscript only analyzes the T cell zone. As the T cells are only migrating in the T cell zone, taking into account the conduit network based on WGA staining of GCs and B cell follicles would not be appropriate and in fact, may present inaccurate information about the conduit network in the T cell zone. Thus, it is important to specify precisely what zones and what stainings are used in this manuscript to analyze LN networks in this manuscript. Additionally, it will be important to use additional networks, such as the T cell zone stainings for FRCs published in Novkovic Plos Biol 2016 to validate the LNCN networks and not rely on only a single lymph node.

One of the major conclusions from the manuscript is that T cell motility is random. This random motility has already been shown previously by analysis of experimental T cell tracks as well as computationally. In the discussion, it was not clearly indicated how using Cheeger value sheds new light on how LN connectivity affects T cell search for lymph nodes. It will be important to discuss the novel contribution of these new analysis on T cell behavior and how it likely impacts overall immune responses.

Reviewer #2: report on manuscript D-22-01540

First and foremost, I want to apologize with the authors for the delay in the delivery of my report. Unfortunately, several issues (international relocation, extra caring duties, and administrative deadlines) have reduced considerably my ability to deliver this report on time.

In this manuscript, the authors present a characterization of the lymph node conduits network (hereafter LNCN) to assess whether its structure optimizes exploration by making some regions more accessible than others. To achieve such a goal, the authors compare the properties (the Cheeger index, $h$, and the enter -- $p_{in}$ -- and exit -- $p_{out}$ -- probabilities) of the communities of the LNCN dual network existing in the so-called diffusion space. The diffusion space is the space where the nodes' relative positions quantify how often two nodes tend to belong to the path made by distinct random walkers. The authors extract the communities of the dual LNCN network by using the k-means algorithm on the position of the nodes in the diffusion space. Finally, to measure the diffusion properties of the LNCN network, the authors make a comparison of the values of $h$, $p_{in}$, and $p_{out}$ computed on the latter, and the same indicators computed for several benchmark networks. The authors have found that although the properties of the LNCN network display a certain spatial regularity, some of its regions seems to promote exploration more.

The application of network science and stochastic processes tools for the analysis of biological systems has proven very useful for scientists working in biological complex systems. The present manuscript poses an interesting question but, I have several concerns about the methodology used to address it. Therefore, my recommendation is that the authors address these concerns and submit a revised version of the manuscript. In the following, I will provide a more detailed explanation of the manuscript's main flaws, together with some suggestions to help the authors addressing them.

#### MAIN ISSUES ####

-- One of the manuscript's pivotal points is to group nodes not according to their topological features (i.e., physical connections) but, rather, according to their diffusion properties. As the goal of the manuscript is to assess how optimized for exploration the structure of the LNCN is, using the diffusion space seems quite natural. However, from a network perspective I have found the decision to project the nodes onto a diffusion space first, and their assignment to a fixed number of communities -- via the $k$-mean algorithm, -- then, quite intricate (maybe I have missed something). There exist, in fact, several methods based on random walks to extract the community structure of a network which perform really well in case of "flow networks" like the LNCN. The most famous of these methods is the so-called Infomap algorithm:

https://www.pnas.org/doi/full/10.1073/pnas.0706851105

One of its authors (Martin Rosvall) maintains a very nice website where, beside information on the algorithm, one can run directly the infomap on its own network

https://www.mapequation.org/

I strongly recommend to, at least, replicate the analysis proposed in this manuscript using the community structure extracted with Infomap directly on the structural network. This step will remove the problem of arbitrarily fixing the number of communities (which does not make much sense to me), as well as reduce the biases on the interpretation of the results found.

-- Another pivotal point of the manuscript is to compare the diffusion properties of the LNCN network with some benchmark models. I am quite sympathetic with this approach but not entirely satisfied with the criteria used to select which these benchmarks and their properties. More specifically:

** The first benchmark I would have considered, would have been a randomized counterpart of the LNCN based on the so-called configuration model. Basically, one keeps the number of nodes, the number of edges, and the degree sequence (i.e., the list of the degrees of each node) of the original network and rewire the connections uniformly at random. There exist a quite efficient algorithm (beside the one available in NetworkX) which generates such type of networks:

https://epubs.siam.org/doi/10.1137/16M1087175

** One criterion which is usually kept in consideration when generating benchmark networks is to keep both the number of nodes, $N$, and the average degree, $\\langle k \\rangle$, equal to those of the original network. The reason is that usually many features (for instance, related to dynamics like diffusion) are intimately related with the network's size and connectivity. The LNCN network has $N = 192386$ and $\\langle k \\rangle = 2.86$. However, the benchmark networks do not preserve always these quantities as some networks are considerably smaller $N = 2050$ and more connected $\\langle k \\rangle \\simeq 960$ like in the case of the Random Geometric Graph. Overall, my sensation is that the comparison between the LNCN and the benchmarks is not made under "fair conditions."

** On a less severe extent, it would be good to quantify how "modular" the networks (LNCN and the benchmarks) are. To do so, one could compute for instance the so-called modularity indicator (NetworkX has a built-in function doing it):

10.1016/j.physrep.2009.11.002 (and references therein)

-- The manuscript does not provide a strong "biological interpretation" of the results found. Perhaps I have missed something but, after reading the manuscript, one question that came to my mind was: "what have we learned from the biological point of view about the LNCN network?" Said in other terms: why it is important to study the diffusion properties of the LNCN, and why what has been found is important/interesting for scientists working with the LNCN? I have also found strange the absence of mentions to already existing results in the Discussion section, as if the results found in this manuscript are disconnected from the existing knowledge on the same topic. Perhaps these points are already covered in the manuscript, but I suggest to make them "shine" more.

-- The presentation style needs to be improved. I believe that a good portion of the "Materials and Methods" section should be moved at the end of the manuscript. The reason is that the current manuscript's organization forces the reader to jump many times back and forth between the methods and the appendices, thus disrupting the reading "flow." The figures are quite poor as they are very tiny (and, sometimes, almost unreadable) and using a color scheme which does not render well when the manuscript is printed in b/w. Concerning this point, there is a flurry of activity around the proper use of color in scientific communication, as explained very well in this nice paper:

https://www.nature.com/articles/s41467-020-19160-7

In general, I recommend to improve the figures by:

** Avoiding using colorscales that are not well rendered in b/w or grayscale (e.g., the jet or rainbow one).

** Enlarge the figures such that details like the axis and ticks labels in the plots are well readable.

** Avoiding using 3D display and, whenever possible, favour heatmaps in their place.

** In line plots [like those of Fig. 3(k-l)] try to use line styles that allow to discriminate distinct datasets even when the figure is displayed in grayscale.

** Add legends to plots if those improve their interpretability.

** Avoiding using colors like yellow, cyan, and light green on a white background as their contrast is not very high and becomes hard to distinguish them.

** Specify always what is displayed on each axis [for instance, improve Fig. S2(a)].

** Condensate plots whenever possible [for instance, Fig. S4 could be replaced by a single line plot with many lines, each accounting for a distinct case].

-- In the data and code availability statement, the authors have stated that "All relevant data are available on Zenodo, and code on github repository." However, I have found no trace of any reference to the data/code in the manuscript. Please, fix this problem.

#### MINOR ISSUES ####

-- In the abstract, the LNCN acronym is used without introducing it. Either use the explicit form and remove the acronym or introduce it properly.

-- I believe that the equation appearing in panel E of Fig. 1 is completely useless as the reader is unable to understand its meaning without reading the whole manuscript. I suggest to remove it.

-- On page 5, Equation (2), the sum runs over $y$ but it is not clear what $y$ is.

-- Page 5 line 127, the symbol to describe the number of nodes is $N$ not $n$. Please correct it.

-- The right way to spell Euclidean is with capital E. Please correct it (there are several instances across the text).

-- Page 6 line 165, there is a missing ) in the equation describing the Cheeger coefficient.

-- I suggest to add some reference to the Cheeger mixing coefficient (I had never heard of it insofar).

-- Page 9 line 259. Perhaps I am mistaken but the correct spelling of "sparcity" is "sparsity".

-- Page 12 line 361. There is an extra ) in the relation of $p_{out}$.

-- Page 12 line 384. Please provide a definition of the diffusion distance.

-- Page 15 Fig. 6 caption. The concept of "biological scan time" is not very clear. Please improve the clarity.

-- Page 17 line 562. Please add a citation to the SciPy library.

-- Page 18 line 573. There is a mention to Eq. (9) of the main manuscript but there are only two equations in the main manuscript. Please clarify this point.

-- The captions and the figures must appear in the same place. However, this is not the case for all the figures of the Appendices. Please fix this.

-- Page 18 line 602. The indices of the first term in the left hand side of the equation displayed do not seem correct to me (there are too many $i$). Perhaps I am wrong but, please, check them.

-- Page 18 line 603. There is a mistake with the URL displayed there. The correct one is:

https://www.stat.berkeley.edu/~aldous/RWG/Chap4.pdf

-- Page 20 line 657. Please state clearly with respect to what the maximum and minimum are computed.

-- Page 23 line 719. There is a typo (the repeated twice).

-- Bibliography.

** Ref. 1 . Please provide more information on this entry. To the best of my understanding it refers to the following book:

https://pascal-francis.inist.fr/vibad/index.php?action=getRecordDetail&idt=PASCALZOOLINEINRA8050247635

** Ref. 17 . The correct reference for this manuscript is:

https://dl.acm.org/doi/10.5555/2976248.2976368

Please, update the information.

** Ref. 19 . Please complete the information about this book (missing publisher and edition).

** Ref. 20 . Please complete the information about this PhD Thesis. I have found something at:

https://www.theses.fr/2020UNIP7037

** Ref. 34 seems a duplicate of Ref. 22 . Please check and, eventually, remove the duplicate entry.

**Have the authors made all data and (if applicable) computational code underlying the findings in their manuscript fully available?**

Reviewer #1: **No: **The biological data used is not clearly defined.

Reviewer #2: **No: **please check my report

PLOS authors have the option to publish the peer review history of their article (what does this mean?). If published, this will include your full peer review and any attached files.

Reviewer #1: **Yes: **Judy Cannon

Reviewer #2: No
---

## [Decision Letter · Decision Letter 1]

25 Apr 2023

Dear PhD Song,

Thank you very much for submitting your manuscript "Random walk informed heterogeneities detection reveals how the lymph node conduits network influences T cells collective exploration behavior" for consideration at PLOS Computational Biology. As with all papers reviewed by the journal, your manuscript was reviewed by members of the editorial board and by several independent reviewers. The reviewers appreciated the attention to an important topic. Based on the reviews, we are plan to accept this manuscript for publication pending minor revisions to address the points of Reviewer 2. 

Sincerely,

James R. Faeder

Academic Editor

PLOS Computational Biology

Mark Alber

Section Editor

PLOS Computational Biology

Reviewer's Responses to Questions

**Comments to the Authors:**

Reviewer #1: This revised manuscript addressed all my concerns. While the use of only 1 biological sample is a major limitation, the tools developed and used to analyze the full lymph node is useful and novel, and can be used as more biological samples become available. This concern is also addressed by the authors in the manuscript.

Reviewer #2: Second report on manuscript PCOMPBIOL-D-22-01540R1

After reading first the authors' reply to the comments made during the previous round of review, and then the revised version of the manuscript, I have come to the conclusion that the authors have addressed satisfactorily to both reviewers' criticisms. The manuscript is now more solid, clear, and the results are properly highlighted and appealing for scientists working in computational biology (and not), and I thank the authors for their efforts in accommodating the reviewers' suggestions.

In light of this, I can now recommend the acceptance of this manuscript for its publication in PLoS Computational Biology. I have only a couple of minor (cosmetic) remarks that the authors should take care of before final publication.

### MINOR ISSUES ###

-- When introducing the random network benchmark, I believe that beside Ref. 26 it would be advisable to cite the work of Bender and Canfield in which the configuration model was introduced

10.1016/0097-3165(78)90059-6

-- The main idea behind the Cheeger mixing index is very similar to the concept of "participation coefficient" used in the following work of Guimera et al.:

10.1038/nature03288

-- Page 6 line 169. The manuscript claims that high values of Cheeger mixing denote that "nodes that belong to the same diffusion community are scattered across the network". However, I believe it is more appropriate to say that high values of Cheeger mixing denote that nodes belonging to the same community do not form contiguous groups as scattered suggests that they are not physically close, which is not necessarily the case.

-- Clarity and readability:

** Page 4 line 128. A simpler (and computationally more efficient) way to compute $\\phi_{j0}$ is by replacing the sum at the denominator with a constant equal to twice the total number of edges.

** Page 6 line 138. I suggest to remove the dot in the expression of the diffusion distance $D$ as, usually, such a symbol denotes the scalar product and can, thus, generate confusion in the reader.

** Page 6 line 148. When mentioning the k-means algorithm, I believe it would be good to cite the paper where it was introduced:

10.1109/TIT.1982.1056489

** Page 7 line 197. Please replace "don't" with "does not".

** Page 8 line 227. Please replace "gaussian" with "Gaussian".

** Page 8 line 251. The symbol "t" in the phrase "when t grows higher" has to be rendered in mathematical font as it denotes the variable $t$.

** Page 9 line 263. Although it is not wrong, I believe it is better to talk about "Monte Carlo simulation" rather than "agent-based model" to indicate the operative implementation of a random walk.

** Page 11 lines 286-287. When describing the implication of the term "small-world" perhaps it would be good to clarify that the clustering coefficient refers to the propensity to form triangles as the use of the term cluster in the rest of the manuscript denotes a different thing and, therefore, might generate confusion in a reader not very familiar with networks. Also, I suggest to cite the work of Watts and Strogatz on the small-world effect (Ref. 21 of the bibliography)

** Appendix S1. I suspect that the phrase "which as the same as" should be "which are the same as". Please, check.

**Have the authors made all data and (if applicable) computational code underlying the findings in their manuscript fully available?**

Reviewer #1: Yes

Reviewer #2: Yes

PLOS authors have the option to publish the peer review history of their article (what does this mean?). If published, this will include your full peer review and any attached files.

Reviewer #1: **Yes: **Judy L Cannon

Reviewer #2: No

Figure Files:

Data Requirements:

Reproducibility:

References:

---

## [Editor Report · Decision Letter 2]

9 May 2023

Dear PhD Song,

We are pleased to inform you that your manuscript 'Random walk informed heterogeneity detection reveals how the lymph node conduit network influences T cells collective exploration behavior' has been provisionally accepted for publication in PLOS Computational Biology.

Best regards,

James R. Faeder

Academic Editor

PLOS Computational Biology

Mark Alber

Section Editor

PLOS Computational Biology

---

## [Editor Report · Acceptance letter]

19 May 2023

PCOMPBIOL-D-22-01540R2 

Random walk informed heterogeneity detection reveals how the lymph node conduit network influences T cells collective exploration behavior

Dear Dr Song,

I am pleased to inform you that your manuscript has been formally accepted for publication in PLOS Computational Biology. Your manuscript is now with our production department and you will be notified of the publication date in due course.

With kind regards,

Anita Estes
